# Age-Related In Vivo Structural Changes in the Male Mouse Olfactory Bulb and Their Correlation with Olfactory-Driven Behavior

**DOI:** 10.3390/biology12030381

**Published:** 2023-02-28

**Authors:** Pietro Bontempi, Maria Jimena Ricatti, Marco Sandri, Elena Nicolato, Carla Mucignat-Caretta, Carlo Zancanaro

**Affiliations:** 1Department of Computer Science, University of Verona, Strada Le Grazie 15, I-37134 Verona, Italy; 2Department of Neuroscience Biomedicine & Movement Sciences, University of Verona, Strada Le Grazie 8, I-37134 Verona, Italy; 3Department of Molecular Medicine, University of Padova, Via Marzolo 3, I-35131 Padova, Italy; 4Centro Piattaforme Tecnologiche, University of Verona, Strada Le Grazie 8, I-37134 Verona, Italy

**Keywords:** magnetic resonance imaging, behavioral test, diffusion tensor imaging, aging

## Abstract

**Simple Summary:**

In humans and mice, the olfactory system is linked to areas of the brain that modulate behavior. Malfunctions of the olfactory system were described in many neurological and psychiatric diseases, including Alzheimer’s and Parkinson’s disease, depression, and mood disorders. Some of these diseases are common in aging persons; hence, it is relevant to know how the olfactory bulb, the first olfactory center in the brain, changes with age in its structure, and how these changes may be related to behavioral modifications. Since most of the experimental work is performed in mice, we studied young and elderly mice with a battery of behavioral tests to describe the differences in motor, olfactory, cognitive, and emotional performance. Then, mice underwent magnetic resonance imaging to describe the differences between olfactory bulbs of young and elderly mice. Lastly, we selected the behavioral variables more predictive of the differences between young and elderly mice and correlated them with the most predictive magnetic resonance variables. Elderly mice were less scared than young mice by new environments, and their olfactory bulb differed in two structural variables, which correlated with three anxiety measures. These data suggest a new direction for human aging studies, on the link between the olfactory bulb and behavior.

**Abstract:**

Olfactory areas in mammalian brains are linked to centers that modulate behavior. During aging, sensitivity to odors decreases and structural changes are described in olfactory areas. We explored, in two groups of male mice (young and elderly, 6 and 19 months old, respectively), the link between the changes in olfactory bulb structure, detected with magnetic resonance imaging, and behavioral changes in a battery of tests on motor, olfactory, cognitive performance, and emotional reactivity. The behavioral pattern of elderly mice appears less anxious, being less scared by new situations. Additionally, the olfactory bulb of young and elderly mice differed in two variables derived from magnetic resonance imaging (fractional anisotropy and T2 maps). A random forest approach allowed to select the variables most predictive of the differences between young and elderly mice, and correlations were found between three behavioral variables indicative of anxious behavior and the two magnetic resonance variables mentioned above. These data suggest that in the living mouse, it is possible to describe co-occurring age-related behavioral and structural changes in the olfactory bulb. These data serve as a basis for studies on normal and pathological aging in the mouse, but also open new opportunities for in vivo human aging studies.

## 1. Introduction

Aging is a complex process involving all body systems. Above 40 years of age, increasing functional olfactory deficit has been consistently found in humans (presbyosmia) [1,2,3,4,5]. However, the structural and functional bases for such a deficit are not completely elucidated.

In mammals, the olfactory bulb (OB) is the first relay structure in the olfactory system; here, axons from sensory neurons in the main olfactory and vomeronasal epithelium of the nasal cavity synapse with tufted and mitral cells, sending, in turn, axons off to the olfactory cortex [6]. Further, the adult OB receives differentiating cells from the subventricular zone, which are integrated as new neurons into existing neural circuits throughout life [7,8]. This implies that the OB maintains a rather juvenile environment with ageing. Nevertheless, the human OB volume has been shown to decline with age in autoptic material [9] and in vivo [1]. Investigation of the mouse brain cannot substitute that in the human; however, working on mice has the advantage of, e.g., limited inter-individual variability, because mouse social activities are simpler than those of humans, and a more homogeneous genetic background. Further, a remarkable characteristic of the olfactory system is that its overall organization is well-conserved among species [10], which confers a translational potential to mouse investigations in this field.

To date, a satisfactory understanding of age-related changes in the olfactory system of the mouse is lacking. Some changes (especially neuronal loss) have been reported in the OB of ageing rodents by histology in early work [11,12,13]. More recently, the authors of [14] and [15] respectively reported that in the rat OB, the neuronal cell number starts to decline as early as three months of age and that ageing was associated with a decrease in the axonal transport rate and bulk transport of material in the olfactory tract in vivo. In the mouse OB, the authors of [16] showed that the number of neuronal cells presents a bell-shaped age dependence with an increase up to 13 months of age and a significant reduction in the 25-month-old individuals. At variance with the above findings, the authors of [17] showed a lack of gross changes in the OB of mice up to 24 months of age with maintenance of the laminar and cellular organization associated with subtle alterations found in glomerular synaptic circuits in the presence of a similar bulb volume. As far as OB volume is concerned, conflicting results were found [13,18,19,20]. As a consequence, further investigation is needed to assess the structural condition of the rodent OB during aging.

The olfactory system extends its role well beyond the sensory processing of external chemicals in both macrosmatic and microsmatic animals [21,22]. Olfactory projection areas are deeply intermingled with systems that modulate behavior, such as the limbic circuit. In the mouse, olfactory inputs may deeply shape behavior towards both environmental and conspecific stimuli, by modulating food search and ingestion, intra- and inter-specific interactions, exploration, and by driving anxiety responses [23,24].

The olfactory system (including secondary and tertiary projection areas) is also involved in different neurological diseases, including epilepsy [25], autism [26], and Parkinson’s and Alzheimer’s diseases [27,28], with a special focus on age-related dementia [29,30]. Structural differences have been detected in ageing mice in olfactory projection areas such as the OB [31], the nucleus of the lateral olfactory tract [32], and the amygdala [33]. Overall, the data reported above show that it should be interesting to explore structural changes in relation to behavioral modulation during ageing.

Magnetic resonance imaging (MRI) is a powerful, non-invasive tool to investigate age-related changes in the olfaction-related structures of the brain in vivo [34]. While its resolution is lower than histology, MRI yields important structural and functional data in intact organisms without preparation artefacts (e.g., shrinkage) and can be performed in both humans and experimental animals, thereby supporting translational research. MRI has been used to investigate the structure and function of the olfactory system [15,35,36,37,38,39,40,41] and has been shown amenable to studying the mouse olfactory bulb in detail [41]. To the best of our knowledge, MRI investigation of age-related changes in the OB in rodents is lacking.

In the present in vivo cross-sectional study, we investigated the size and structural characteristics of the OB at two different ages by means of MRI as well as the animals’ behavioral pattern by means of an extensive battery of behavioral tests in the same mice. The aim was two-fold: first, to verify the presence of age-related changes in OB as well as behavioral changes, and second, to assess the association of MRI variables with behavior, in order to link structural and behavioral modifications.

## 2. Materials and Methods

### 2.1. Animals

Male CD-1 mice, 6 or 19 months old, were reared at in-house animal facilities. Mice were maintained in groups of 4–6 in standard cages, at 24 ± 1 °C, on a 12:12 light regimen, with lights on at 6.00 a.m. Mice were reared with both parents and caged in same-gender groups from weaning, with mouse food pellets and water always available. Experiments were approved by the Italian Ministry of Health (number 43F3E.0) and conducted according to European regulations on animal research (2010/63/EU).

### 2.2. Behavioral Tests

A battery of tests was administered to measure motor performance, emotional reactivity to environmental stimuli in different challenging conditions, olfactory function, intra- and inter-specific interactions, and memory. Tests were administered on different days.

#### 2.2.1. Cord Test

To evaluate the motor performance, a cord (50 cm long) was placed over a box, 10 cm from the bottom, which was covered with soft material. Mice should take the cord in the middle with forepaws and cross to either side (maximum time 180 s). The latency before falling or the time to reach the end of the cord was recorded.

#### 2.2.2. Open-Field Test

The mouse locomotion and drive to exploration were measured in a new environment. The mouse was introduced in a plastic cage (55 × 33 × 20 cm) for 180 s. The test was digitally recorded and the travelled distance, resting time, and rearing on the walls were quantified with SMART 2.5 (2 Biological Instrument, Varese, Italy). Additionally, the number of fecal pellets and urine drops was recorded, as indexes of autonomic activation.

#### 2.2.3. Swim Test

The motor performance and emotional reactivity was measured in a new/unknown environment by putting the mouse at the center of a plastic pool (55 × 35 × 30 cm) filled with 20 cm-deep water (25 °C). Their behavior was recorded for 180 s. The swimming distance and resting time were analyzed with SMART 2.5 software (2 Biological Instrument, Varese, Italy).

#### 2.2.4. Light Avoidance

To evaluate preference for a dark environment, the mouse was put in a plastic cage divided into two compartments (each 21 × 27 × 15 cm) by a plastic wall with a 4 × 3.5 cm opening. One compartment was white and open to allow illumination from above. The other side was painted black and closed on the top. The mouse was firstly put in the dark compartment and the time for the mouse to go to the light compartment was recorded (dark-to-light time, DL1). The mouse was then picked up and rested in the home cage. Then, it was put back into the light compartment and the time until the mouse entered the dark side was recorded (light-to-dark time, LD). Then, the time for the mouse to freely return to the light compartment was recorded (DL2). The maximum time for each trial was 180 s. If a mouse failed to respond within this time, it was assigned a score of 180.

#### 2.2.5. Olfactory Test 1: Food Finding

To evaluate olfactory functions, mice were deprived of food overnight, then given 300 s to habituate and explore a clean cage (42 × 26 × 13 cm), with a food pellet (1.5 × 3 cm) buried under the sawdust. They were then returned to their home cage, and the test was repeated recording the time (in seconds) to complete the task, until mice dug out the pellet and started to bite. Then, the test was repeated placing the food pellet in a visible position over the sawdust, to control for motivation to eat [42].

#### 2.2.6. Olfactory Test 2: Olfactory Preference Test

To evaluate the mice exploration of familiar (same species male urine) and unfamiliar olfactory stimuli (linalool), mice were put in a cage (45 × 24 × 20 cm) with Benchguard paper in the bottom for three trials, lasting 180 s each. An area of 10 × 10 cm was marked 15 cm apart from each shorter side of the cage, containing two drops of stimuli. The stimuli were: water on both sides (first trial—control condition without odorants), water in one square and linalool in the other (second trial—odor condition), and water and familiar urine (third trial—urine condition). The time until the first sniff (latency) and the number of sniffs were recorded. Travelled distance in each side of the cage was measured with SMART 2.5 software.

#### 2.2.7. Pole Test

The pole test evaluates motor performance, but by repeating it on consecutive days it allows to evaluate learning. Mice were placed head-up on a pole (50 × 1.5 cm) covered with gauze to assist grasping. The time required to turn down and reach the floor was recorded (maximum 180 s). This test was repeated on 4 consecutive days.

#### 2.2.8. Intraspecific Intruder Test

The intruder test evaluates intra-species aggressiveness towards unrelated mice of the same strain, sex, and age, after placing the intruder mouse in the home-cage of the mouse. The latency time to the first aggressive attack was recorded for 30 min.

#### 2.2.9. Predation Test

The predatory behavior was evaluated by placing each mouse in a plastic cage (25 × 15 × 13 cm) containing an earthworm (*Lumbricus terrestris*). The latency time to the first attack was recorded in 30 min.

### 2.3. Magnetic Resonance Imaging

For MRI, mice were anesthetized by inhalation of a mixture of air and O_2_ containing 0.5–1% isoflurane and placed in a prone position with their head in stereotactic position. Images were acquired using a Biospec Tomograph (Bruker, Karlsruhe, Germany) equipped with a 4.7 T, 33 cm bore horizontal magnet (Oxford Ltd., Oxford, UK). A double-coil configuration was used: the excitation radiofrequency pulses were applied through a 7.2 cm birdcage volume coil, while the signal was received through a 2-channel surface coil optimized for the mouse brain (Bruker). After a sagittal scout image, 12 contiguous 0.5 mm-thick slices were acquired through the whole OB using a RARE T2-weighted sequence with TR = 5000 ms, TE = 76 ms, FOV (field of view) = 2.0 cm^2^, NEX (number of average) = 10, and matrix size = 192 × 192, corresponding to an in-plane resolution of 104 × 104 μm^2^. These images were used for the calculation of OB volume. Then, five contiguous 1 mm-thick slices of the whole OB were acquired using a multi-echo spin-echo sequence (MSME) with matrix size = 256 × 192, FOV = 2.5 × 2.5 cm^2^, corresponding to an in-plane resolution of 100 × 132 μm^2^, NEX = 1, 10 echoes, TR of 2611 ms, and TE ranging from 20 to 200 ms. These images were used for generating quantitative T2 mapping [43].

#### 2.3.1. Cerebral Blood Volume

Fast low-angle shot magnetic resonance imaging (FLASH) sequences (TR/TE = 350/15 ms, flip angle = 30°, number of slices = 5, slice thickness = 1 mm, matrix size = 256 × 192, FOV = 2.0 × 2.0 cm^2^, corresponding to an in-plane resolution of 78 × 104 μm^2^, NEX = 2) of the OB were acquired before and two minutes after intravenous administration of Endorem^®^, kindly supplied by Guerbet (Villepinte, France), in order to obtain maps of relative cerebral blood volume (rCBV) [44].

#### 2.3.2. Diffusion Tensor Imaging

Diffusion tensor imaging (DTI) is a MR modality based on the observation of water molecule diffusivity in the brain. DTI is especially employed to characterize the orientation and integrity of the white matter [45,46]. Diffusion tensor imaging (DTI) is a neuroimaging sequence sensitive to microstructural and cellular changes occurring prior to volumetric changes [47,48]. DTI images were acquired with an Echo Planar Imaging (EPI) sequence optimized for the mouse olfactory bulb. Imaging parameters were set as follows: TR = 3000 ms, TE = 36 ms, FOV = 2 × 2 cm^2^, matrix size = 128 × 128, corresponding to an in-plane resolution of 156 × 156 μm^2^, NEX = 10, and 12 transversal 0.5 mm-thick slices. Diffusion images were acquired in 12 non-collinear directions and 2 b-values were used (0 and 750 s/mm^2^). DTI-derived maps, fractional anisotropy (FA), apparent diffusion coefficient (ADC), axial diffusivity (AD), and radial diffusivity (RD) were calculated with Paravison 5.1 (Bruker, Karlsruhe, Germany). DTI data were then analyzed by using the FSL software package (http://fsl.fmrib.ox.ac.uk/fsl/fslwiki/ accessed on 20 November 2020). EPI images were converted into NIFTI file format, preserving the original scanner orientation but with a 10× scale factor on the voxel size to be FSL-compliant for subsequent processing. To evaluate DTI-derived parameters, a mask, covering the whole olfactory bulb, was manually traced on each subject b0 image with FSLVIEW. The mean values of the considered parameters (FA, ADC, AD, or RD) were then extracted.

#### 2.3.3. OB Volume

Semi-automatic analysis of the OB cross-sectional area was performed using the software Paravision 5.1 (Bruker, Karlsruhe, Germany) on T2-weighted MR images. Total OB volume was calculated by one operator according to the following formula: V(obj) = t × Σa(s), where V is the total OB volume, t is the slice thickness (0.5 mm), and Σa(s) is the sum of the areas of all cross-sections of the object. The intra-observer intraclass correlation coefficient (Cronbach’s alpha) for repeated measurements was 0.998.

### 2.4. Statistical Analysis

Data were generally non-normally distributed according to the Shapiro–Wilk test; accordingly, nonparametric statistics was used throughout the analysis. The Mann–Whitney test was used to compare variables between the two groups of mice. Data are presented as the median (interquartile range). The Friedman test was used to compare performance across the four consecutive days of the pole test. In case of significance, the Mann–Whitney test was used for comparison of the two age groups of mice on the same day. Pairwise correlation analysis was performed calculating the Spearman rho for continuous variables. The strength of the correlation coefficient was considered small (0–0.30), moderate (0.31–0.49), large (0.50–0.69), very large (0.70–0.89), and almost perfect (0.90–1), as per [49]. We also performed a joint multivariable analysis in order to identify the set of variables that shows the strongest differences between the age groups, alone or in mutual interaction. This analysis was performed by estimating permutation variable importance measures (VIMs) using random forests [50]. VIMs are a measure of the difference between groups of the distribution of each variable, individually as well as in multivariate interactions with other variables. Using the method of [51], we also estimated the confidence intervals (CIs) of VIMs and selected the subset of variables with CIs not intersecting the zero line.

A *p*-value equal to or less than 0.05 was considered statistically significant. Statistical analysis was carried out with the IBM-SPSS (v.25) statistical package or the R package v. 4.2.2 (R Core Team, 2014) with the Random Forest SRC package v. 3.1.1, and graphs were generated with Prism9 (Version 9.4.1).

## 3. Results

Complete, valid data were obtained for all MRI and behavioral variables in a minimum of 22 and 19 mice aged 6 and 19 months, respectively, and the actual number of mice in the analyses is reported in the tables. Young and elderly mice had a similar body mass (48.50 (6.50) g vs. 49.75 (5.63) g; *p* = 0.183) (Figure 1).

### 3.1. MRI

Table 1 shows the comparison of MRI variables in young and elderly mice (Mann–Whitney test). Figure 2 shows representative MRI images of an adult CD-1 mouse head (Figure 2A) showing the area used for OB morphometry (Figure 2B) and a T2 map (Figure 2C) as well as an FA map (Figure 2D).

T2 and bulb volume values were lower in elderly than young mice, with the difference being statistically significant (*p* < 0.001) for the former only. Analysis of DTI parameters showed that FA was significantly higher in elderly mice (*p* = 0.004), whereas ADC, AD, and RD were similar in the two age groups (*p* > 0.05). rCBV was also similar in the two age groups (*p* = 0.285).

Overall, we showed that some in vivo measurable variables differ between elderly and young mice.

### 3.2. Behavioral Tests

The results of behavioral tests in the two age groups are presented in Table 2 and Figure 3 and Figure 4.

#### 3.2.1. Motor Performance

The cord test showed that elderly mice fell significantly earlier than young mice from the cord (*p* = 0.038, Figure 3A). In the open-field test, no statistically significant difference was apparent between young and elderly mice, with the exception of the number of rearing on the walls, which was higher in young mice (*p* = 0.007, Figure 3B). In the swim test, the elderly mice swam a longer distance than young mice (*p* = 0.007, Figure 3C), while resting time did not differ in the two age groups. As for the motor component of the light avoidance test (Figure 3D), the time to the first transition from the dark to the light side of the cage (DL1) was similar in the two age groups; however, the time to go from the light to the dark side (LD) was significantly shorter in young mice (*p* < 0.001). Elderly mice were faster than young mice in going from the dark to the light side (DL2, *p* = 0.001). Young mice stayed longer in the dark compartment in both conditions, either placed by the experimenter or voluntarily entering the dark compartment (DL1 > LD and DL2 > LD, *p* < 0.001 in both conditions). However, their DL2 transition was significantly slower than the first one (DL1 < DL2, *p* < 0.001). The elderly mice, instead, performed the same as young mice in DL1, which was significantly longer than the LD transition time (*p* < 0.05). However, DL2 was shorter in elderly than young mice (*p* = 0.001) and did not differ from the LD transition time.

#### 3.2.2. Olfactory Function

No statistically significant difference was apparent between age groups in the food-finding test. Instead, the olfactory preference test highlighted some statistically significant differences between the two age groups. In the control condition, no difference was detected between the performance of young and elderly mice, as well as the within-age group performance on the right and left sides of the cage. When the odorant linalool was present on one side of the cage, no difference was detected between the performance of the two groups in absolute terms; however, when examining the pattern of response within each group, it was found that in both young and elderly mice, the time to reach the odorous area was significantly shorter than the latency to reach the control, non-odor area (*p* < 0.005 for young and *p* < 0.01 for elderly mice, Figure 4A), while the number of sniffs was higher for the odorous stimulus only in young mice (*p* < 0.001, Figure 4B) but not in the elderly (*p* = 0.076). The distance covered by young mice was longer in the odor compartment than in the control compartment (*p* < 0.001, Figure 4C), as also occurred in elderly mice (*p* < 0.05). When urine was present in the cage of young mice, no significant difference emerged between the control and urine-scented compartments in the time to reach each stimulus, in the number of sniffs, and in the travelled distance (Figure 4D–F). On the contrary, elderly mice showed an increased number of sniffs on the scented stimulus (*p* < 0.05) as well as a significantly shorter time to reach the urine-scented stimulus than the control one (*p* < 0.05), also travelling more distance on the scented side (*p* < 0.05).

#### 3.2.3. Cognitive Performance

The time to descend from a pole was statistically significantly different in both the young and elderly groups on the different days. In each group, the time to reach the ground became significantly shorter one day after the other (*p* < 0.01 for all comparisons except in elderly mice on day 3 vs. day 4, *p* < 0.05, Figure 3E). While no difference could be detected between young and elderly mice in the performance of days 1, 2, and 3 (*p* > 0.05), which improved from day to day, on the fourth day the younger mice additionally improved their performance, thus being faster than the elderly mice (*p* = 0.003).

#### 3.2.4. Emotional Reactivity

Emotional reactivity was tested in two tests involving an olfactory component (intraspecific intruder, predation). Only one elderly mouse did not attack the opponent mouse intruder. Additionally, the latency to the first attack to the intruder mouse was not significantly different between young and elderly mice. The behavior towards a prey was, however, different in young and elderly mice, since the attack was faster in younger animals (*p* = 0.020).

#### 3.2.5. Correlation Analysis of MRI and Behavioral Variables

Random forest analysis ranked MRI and behavioral variables according to their relative importance in differentiating young and elderly mice (Figure 5). Using the method reported in [51], the analysis showed that the first nine ranked variables should be considered as the most relevant. Accordingly, correlation analysis of MRI and behavioral variables was carried out for these variables only.

Some most relevant correlations were found. The light-to-dark transition time in the light avoidance test (light avoidance test—LD) negatively correlated with T2 maps (rho = −0.671, *p* < 0.001) and positively with FA (rho = 0.333, *p* = 0.033), while DL2 positively correlated with T2 maps (rho = 0.320, *p* = 0.044). The latency to attack a heterospecific intruder (predation, latency) correlated with FA (rho = 0.403, *p* = 0.009).

In summary, we showed, using a statistically conservative approach, several relationships between in vivo measurable variables and behavioral traits.

## 4. Discussion

### 4.1. MRI Variables in the Two Age Groups

Elderly mice were characterized by a lower T2 and a higher FA compared to their younger counterparts, while bulb volume, rCBV, and other diffusion tensor variables (ADC, AD, RD) were similar in the two age groups. While investigation of the histological/biochemical counterparts of such changes was beyond the scope of this work, the findings are amenable to discussion based on literature data. T2 maps provide information on the status of water in nervous tissue [48,52]. FA is the degree of anisotropic diffusion relative to the overall diffusion, i.e., FA measures the relative diffusion along vs. across fiber tracts, where 0 is completely isotropic diffusion and 1 represents diffusion restricted to one direction, and it is associated with the coherence of white matter and is highly sensitive to microstructural changes [48]. Understanding the relationship between T2 maps and the underlying cyto- and myelo-architecture is not easy because of the number of confounding factors, such as myelin, iron, blood vessels, and structure orientation [53]. For example, the shorter T2 found in the OB in elderly mice could be compatible with an increased amount of myelin in the tissue. The T2 of myelin is rather short (about 20 ms) and a higher presence of myelin would account for a shorter measured global T2 [54]. Interestingly, shorter white matter T2 was associated with better cognitive performance [54], and we showed herein that elderly mice overperformed adult mice in some tasks. Changes in FA have also been attributed to different factors. Once more, increased FA has been associated with increased myelinization [55]; on the other hand, fiber geometry has been advocated as a cause [56]. An increased amount of myelin could therefore justify the measured increase in FA of the OB in elderly mice as well [57], or such difference could be explained by a mere grey matter atrophy of the OB as a result of reduced relative volume. Indeed, we measured MRI metrics on the whole bulb tissue without differentiating between white and grey matter, since voxel size and tissue contrast were too low to drive a reliable tissue segmentation. Hence, a reduction in grey matter volume would account for increased FA and reduced T2 as the relative contribution of white matter increases. In fact, in a healthy mouse brain, FA values are higher in white than grey matter [58] and the T2 of white matter is shorter than grey matter [59].

Overall, the current data highlighted the need for further investigation clarifying the determinants of in vivo age-related MRI changes in the OB.

### 4.2. Behavioral Differences between Young and Elderly Mice

#### 4.2.1. Motor Performance

The elderly mice fell significantly earlier than young mice from the cord, as may be expected by a lower grip strength in the elderly. Similar findings have already been reported in rodents and humans [60]. Motor performance was also explored in the open-field test, which poses an emotional challenge to the mouse, being conducted in a completely new environment. Under these conditions, no difference in motor performance was detected, since only the number of rearing on the walls, an index of emotional reactivity, differed between groups, being higher in young mice. Rearing on the walls is an index of anxiety and consistently increases in different animal models of stress [61,62]. The swim test is slightly more stressful than the open-field test since it requires active movements to stay afloat, while in the open-field test the mouse may eventually stop taking any action. Counterintuitively, in the swim test, the elderly mice swam longer than young mice, showing that the elderly mice had no major motor impairment and were less frightened by the new, relatively hostile environment. Since the resting time did not differ in the two age groups, it may be argued that the elderly mice swam around faster, which rules out major motor impairments in the elderly. Interestingly, an increased latency to the first stop has already been reported in old wild-type mice [42]. Additionally, the light avoidance test has a motor component, since mice may walk freely in the apparatus. The LD time to go from the light to the dark side was faster in young mice, which cannot be entirely explained by a better motor performance in young mice compared to the elderly. Actually, elderly mice were faster than young mice in going from the dark to the light side after having freely entered the dark side (DL2), when they were free to choose their time, suggesting that young mice were more scared by the light compartment and were actively seeking a safer dark environment. Under these conditions, the motor performance clearly reflected the emotional reactivity to environmental challenges. It may be noted that young mice stayed longer in the dark compartment in both conditions, when placed by the experimenter or by voluntarily entering the dark compartment. However, the second transition from dark to light was significantly slower than the first one, suggesting that mice in the first instance were also scared by the manipulation, which affected their risk assessment in driving them to the light compartment. Actually, risk assessment is part of the defense reaction and is modified in behavioral disorders, such as anxiety and depression, in both humans and rodent models [63]. For elderly mice, DL2 was shorter than that of young mice, and did not differ from the LD transition time, suggesting that elderly mice were less scared than young mice by the illuminated environment.

#### 4.2.2. Olfactory Function

The food-finding test, which is sensitive to olfactory impairments [64], indicated a substantial similarity in functional olfactory performance between young and elderly mice. The olfactory preference test, however, highlighted some relevant behavioral differences between the two age groups. The behavior in the control condition was similar, but when the odorant was present on one side of the cage, only in young mice was the number of sniffs higher for the odorous stimulus, and the distance covered was longer in the odorous compartment, which means a faster activity. Since all the other variables did not differ between the two age groups, we may infer that a substantially similar approach was present towards a neutral odorous stimulus in young and elderly mice. However, what happens if the stimulus is meaningful for the species, as it is the case for urine of an otherwise unknown and unrelated adult male mouse? In this case, young mice modified their behavior in the test, since no significant difference emerged between the control and urine-scented compartments in the time to reach each stimulus, in the number of sniffs, or in the travelled distance. Apparently, young mice refrained from approaching the possibly threatening male mouse urine, at variance with the emotionally neutral odor. On the contrary, the behavior of elderly mice was more reminiscent of their approach to a neutral odor, by showing an increased number of sniffs on the scented stimulus and a shorter time to reach the scented side than the control one, while travelling more distance on the scented side. In summary, the olfactory-driven behavior of young and adult mice differs in their approach to differently meaningful stimuli, while no major functional olfactory deficits could be detected in the elderly mice. Noteworthy, age influences the release of male chemosignals, so these molecules may indicate the presence of a rather young male mouse [65]. Moreover, the presence of male mouse urine in the environment may affect the aversion to light [66] and turn male mice sexual behavior into aggression [67].

#### 4.2.3. Cognitive Performance

The performance in the pole test improved from day to day for the first three days in both groups, showing that mice could learn and remember the setup, improving their motor performance accordingly. On the fourth day, young mice additionally improved their performance, while the elderly did not, possibly because they already reached their best performance. In 15-month-old wild-type CD1 mice, a similar learning pattern has also been reported in the same test [68].

#### 4.2.4. Emotional Reactivity

The latency to attack an intruder mouse was not significantly different between young and elderly mice, showing that the cautious approach adopted by young mice towards male mouse urine in the olfactory preference test was overcome by the risk represented by the physically present opponent. A change in behavioral responses after stressors can be detected in both young and aged mice, with increased social avoidance that leads to reduced social interactions [69]. However, young mice were faster than the elderly mice in attacking the prey, further supporting a different risk assessment for triggering aggression in young and elderly mice.

#### 4.2.5. Correlative Analysis of RF-Selected MRI and Behavioral Variables

The negative correlation between LD and T2 maps indicated that a shorter T2 was related to a longer LD transition time, both conditions present more in elderly than in younger mice. LD was also positively correlated to FA, which makes sense because of the opposite trend in T2 maps and FA measures. Interestingly, DL2 time was positively correlated to T2 maps, which strengthened the links between anxious behavior and structural changes in the olfactory bulb. It remains to be elucidated whether a shorter T2 is related to decreased grey or increased white matter in the elderly mice, while increased LD fits the idea of a less anxious behavior. The increased FA values in elderly mice were positively related to the time to attack a prey, which was shorter in young animals. Both LD and predatory attacks were related to anxiety. Noteworthy, lesions of the olfactory bulb induce extremely anxious phenotypes and increased predatory aggression [64]. In the case of normal aging, it appears that the olfactory contribution to behavioral modulation is in the sense of a decreased anxious trait that has a structural counterpart, which may already be apparent in vivo.

## 5. Conclusions

The present data described the differences in behavior between young and elderly mice, according to different behavioral domains that were in the past demonstrated to be related to the olfactory function, as well as the differences in olfactory bulb magnetic resonance imaging. The most relevant differences were then correlated, indicating that it is also possible to explore structural changes in vivo in mice. These data have a translational outlook since a similar approach may also be devised in humans.

## Figures and Tables

**Figure 1 biology-12-00381-f001:**
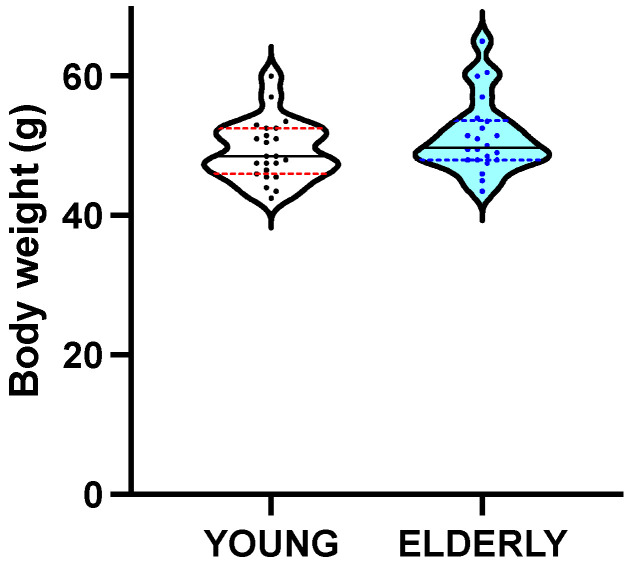
The body weight was similar in the two groups of young and elderly mice. Single data are presented as dots. Black solid line: median, colored dotted lines: interquartile ranges.

**Figure 2 biology-12-00381-f002:**
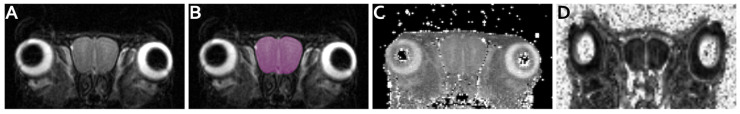
Representative axial MRI pictures of an adult CD-1 mouse head. (**A**) T2-weighted anatomical image. (**B**) The cyan area identifies the bulb region used for volumetric, T2, and FA measurements. (**C**) T2 map. (**D**) FA map.

**Figure 3 biology-12-00381-f003:**
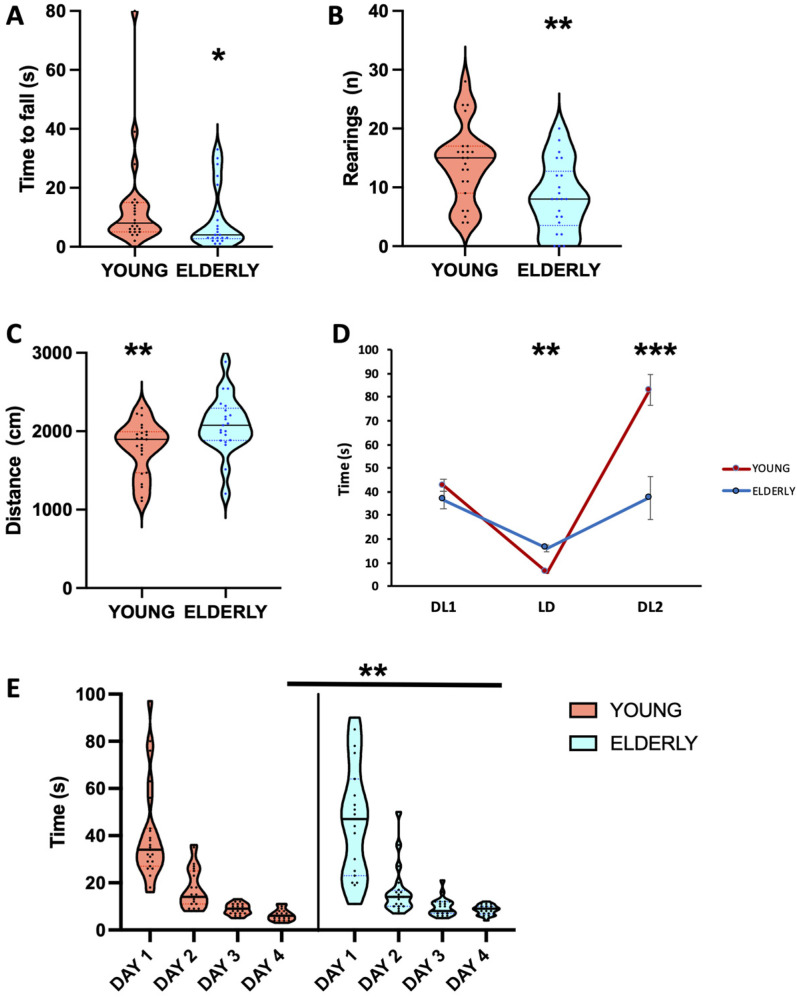
Behavioral performance of young (red) and elderly (light blue) mice. Horizontal solid line: median, colored dotted lines: interquartile range. *: *p* < 0.05, **: *p* < 0.01, ***: *p* < 0.001, young vs. elderly. For detailed statistics, see the main text and Table 2. (**A**) Cord test, time to fall. (**B**) Open-field test, number of rearing on the walls. (**C**) Swim test, distance travelled. (**D**) Light avoidance test, DL1: the first dark-to-light transition time, LD: light-to-dark transition time, and DL2: the second dark-to-light transition time. (**E**) Pole test performance over the four days.

**Figure 4 biology-12-00381-f004:**
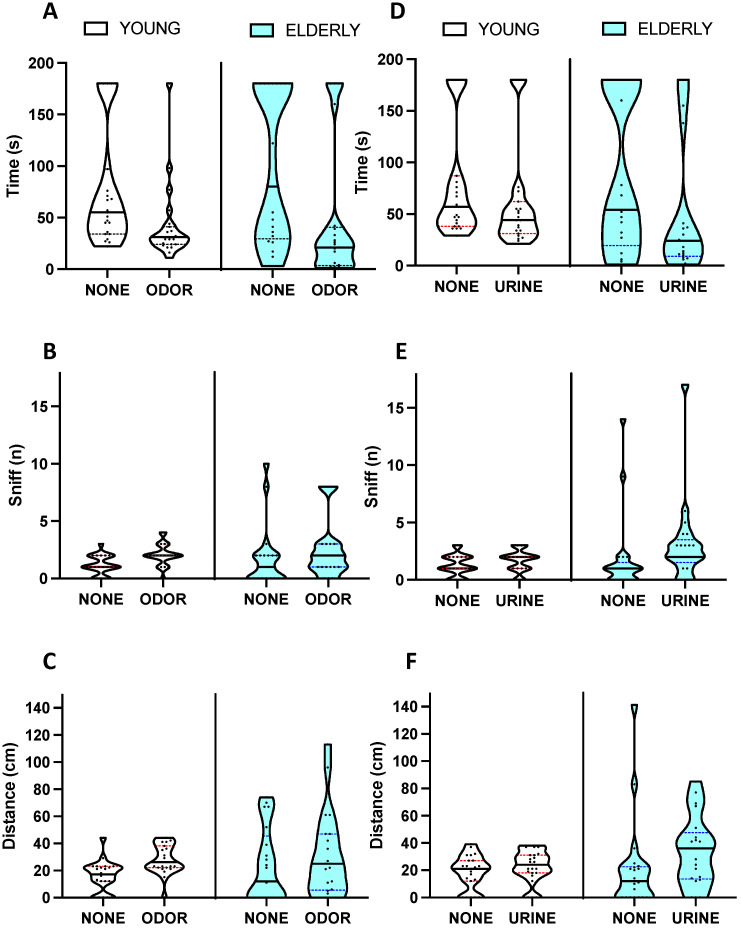
Olfactory preference test performance in young (red) and elderly (light blue) mice. For detailed statistics, see the main text and Table 2. (**A**–**C**) One side of the cage was scented with linalool (ODOR) and compared to the other non-scented side (NONE). (**D**–**F**) One side of the cage was scented with male urine (URINE) and compared to the other non-scented side (NONE). (**A**,**D**) Latency to reach the stimulus area on each side. (**B**,**E**) Number of sniffs to the stimulus area. (**C**,**F**) Distance travelled in each area. Solid line: median. Colored, dotted lines: interquartile range.

**Figure 5 biology-12-00381-f005:**
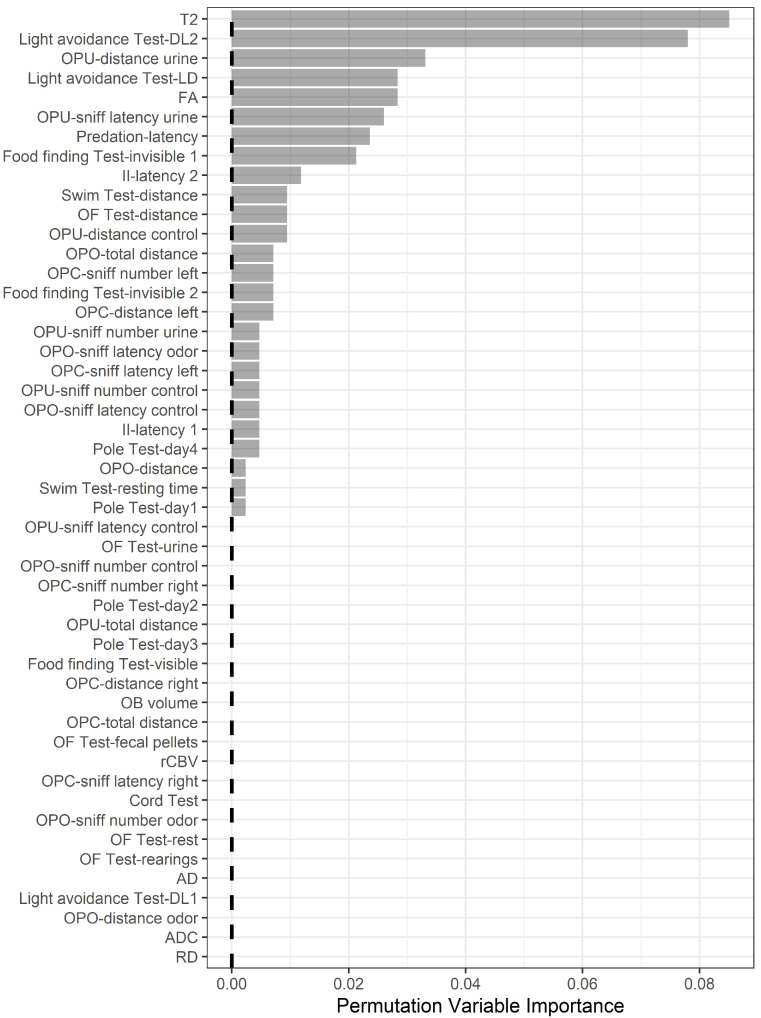
Random forests analysis ranking of MRI and behavioral variables according to their relative importance in differentiating young and elderly mice. OF, open-field; LD, light-to-dark; DL, dark-to-light; OPC, olfactory preference—control; OPO, olfactory preference—odor; OPU, olfactory preference—urine; II, intraspecific intruder; FA, fractional anisotropy; OB, olfactory bulb; AD, axial diffusivity; ADC, apparent diffusion coefficient; RD, radial diffusivity; rCBV, relative cerebral blood volume.

**Table 1 biology-12-00381-t001:** Comparison of MRI variables in the olfactory bulb of 6-month-old (young) and 19-month-old (elderly) mice. Data are median (interquartile range). Mann–Whitney test, significant differences are in bold.

Variable	Young (*n* = 22)	Elderly (*n* = 19)	Z	*p*-Value
Bulb volume (mm^3^)	26.0 (4.0)	25.0 (5.0)	−0.464	0.643
T2 maps (ms)	61.56 (1.80)	57.94 (1.15)	−4.301	**<0.001**
rCBV	0.529 (0.15)	0.555 (0.11)	−1.070	0.285
FA	0.2764 (0.03)	0.3090 (0.07)	−2.902	**0.004**
ADC (×10^−3^) (mm^2^/s)	0.63 (0.02)	0.65 (0.03)	−0.397	0.705
AD (×10^−3^) (mm^2^/s)	0.83 (0.04)	0.88 (0.01)	−1.098	0.272
RD (×10^−3^) (mm^2^/s)	0.54 (0.02)	0.54 (0.06)	−0.837	0.404

rCBV, relative cerebral blood volume; FA, fractional anisotropy; ADC, apparent diffusion coefficient; AD, axial diffusivity; RD, radial diffusivity.

**Table 2 biology-12-00381-t002:** Comparison of behavioral variables in 6-month-old (young) and 19-month-old (elderly) mice. Data are median (interquartile range). Mann–Whitney test, significant differences between the two age groups are in bold.

Test	Variable	Young (*n* = 23)	Elderly (*n* = 21)	Z	*p*-Value
Food finding	Invisible (s)	38.0 (20.0)	37.0 (64.0)	−0.082	0.934
	Visible (s)	21.0 (19.0)	12.0 (30.0)	−1.294	0.196
Intraspecific intruder	Latency 1° attempt (s)	267.0 (249.0)	253.0 (803.0)	−0.590	0.555
Predation	Latency 1° attempt (s)	1800.0 (1278.0)	1800.0 (239.0)	−2.323	**0.020**
Swim	Distance (cm)	1897.0 (522.0)	2077.0 (410.5)	−2.691	**0.007**
	Resting time (s)	26.0 (18.0)	31.0 (29.0)	−0.717	0.473
Pole day 1	Time to descend (s)	34.0 (16.0)	50.0 (55.25)	−1.487	0.137
Pole day 2	Time to descend (s)	14.0 (14.0)	14.0 (11.75)	−0.011	0.991
Pole day 3	Time to descend (s)	9.0 (4.0)	9.0 (5.0)	−0.778	0.437
Pole day 4	Time to descend (s)	6.0 (4.0)	9.0 (3.25)	−3.019	**0.003**
Cord	Time to fall (s)	8.0 (10.0)	4.0 (11.5)	−2.071	**0.038**
Light avoidance	DL1 time (s)	43.0 (20.0)	37.5 (64.5)	−0.352	0.725
	LD time (s)	6.0 (3.0)	17.5 (10.75)	−5.065	**<0.001**
	DL2 time (s)	97.0 (70.0)	20.0 (78.0)	−3.226	**0.001**
Open-Field	Rearings (n)	35.0 (8.0)	8.0 (9.25)	−2.696	**0.007**
	Urine Drops (n)	0.0 (0.0)	0.0 (0.0)	−0.961	0.337
	Fecal pellets (n)	0.0 (1.0)	0.0 (1.25)	−0.103	0.918
	Distance (cm)	959.0 (522.0)	1032.0 (525.75)	−1.056	0.291
	Resting time (s)	44.0 (57.0)	78.0 (76.25)	−1.669	0.095
Olfactory preference- Control	Water (left side): Latency to sniff (s)	31.0 (19.0)	27.0 (40.0)	−1.613	0.107
	Water (left side): Number of sniffs (n)	2.0 (1.0)	2.0 (4.5)	−1.062	0.288
	Water (left side): Distance (cm)	24.0 (17.0)	24.0 (36.5)	−0.789	0.430
	Water (right side): Latency to sniff (s)	36.0 (24.0)	26.0 (26.0)	−2.015	**0.044**
	Water (right side): Number of sniffs (n)	2.0 (2.0)	2.0 (4.5)	−1.176	0.240
	Water (right side): Distance (cm)	25.0 (21.0)	27.0 (44.0)	−1.260	0.208
	Total distance (cm)	978.0 (270.0)	1110.0 (511.0)	−1.821	0.069
Olfactory preference- Odor	Water: Latency to sniff (s)	55.0 (63.0)	80.0 (150.0)	−0.502	0.616
	Water: Number of sniffs (n)	1.0 (1.0)	1.0 (2.0)	−0.233	0.816
	Water: Distance (cm)	17.0 (11.0)	12.0 (45.5)	−0.277	0.820
	Odour: Latency to sniff (s)	31.0 (17.0)	21.0 (37.0)	−1.647	0.100
	Odour: Number of sniffs (n)	2.0 (1.0)	2.0 (2.0)	−0.538	0.591
	Odour: Distance (cm)	26.0 (16.0)	25.0 (41.5)	−0.353	0.724
	Total distance (cm)	1054.0 (266.0)	895.0 (372.5)	−2.103	**0.035**
Olfactory preference- Urine	Water: Latency to sniff (s)	57.0 (49.0)	54.0 8160.5)	−0.261	0.794
	Water: Number of sniffs (n)	1.0 (1.0)	1.0 (1.5)	−0.991	0.322
	Water: Distance (cm)	21.0 (15.0)	12.0 (22.5)	−1.283	0.200
	Urine: Latency to sniff (s)	44.0 (31.0)	24.0 (80.5)	−2.400	**0.016**
	Urine: Number of sniffs (n)	2.0 (1.0)	2.0 (2.0)	−1.920	0.055
	Urine: Distance (cm)	24.0 (13.0)	36.0 (34.0)	−1.389	0.165
	Total distance (cm)	952.0 (192.0)	970.0 (308.5)	−0.576	0.565

## Data Availability

Data are available from the corresponding author upon reasonable request.

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
