# Peer review of "Age-Related In Vivo Structural Changes in the Male Mouse Olfactory Bulb and Their Correlation with Olfactory-Driven Behavior"

_biology, 2023, doi:10.3390/biology12030381_

Round 1
Reviewer 1 Report
In this manuscript, the author explored the link between the difference in olfactory bulb structure in younger and elderly mice. The author evaluated the difference with MRI, and behavioral changes in a battery of tests on motor, olfactory, cognitive, and emotional reactivity. There is no direct evidence to prove the behavioral changes related to mouse olfactory bulb, so the conclusion of this paper is arbitrary. There are still some minor concerns that should be addressed by the author.
Line 298: I recommend the author add a brief conclusion after the MRI result.
Line 307: The author said the P value of the cord test is P<0.05, I recommend the author write the actual P value, P=0.038, which is the same as the number in Table 2. Replace all the P values with the actual P value that showed in Table 2 in this paper.
Line 334: Please replot figure 2 and add a P value and * in each panel.
Line 334: The variation of the YOUNG group in Figure 2A looks too big. please check the P value again if you ignore the biggest value.
Line 387: Please replace “DL, light-to-dark” with “DL, dark-to-light”.
Line 397: Please add a brief conclusion after this part.
Author Response
In this manuscript, the author explored the link between the difference in olfactory bulb structure in younger and elderly mice. The author evaluated the difference with MRI, and behavioral changes in a battery of tests on motor, olfactory, cognitive, and emotional reactivity. There is no direct evidence to prove the behavioral changes related to mouse olfactory bulb, so the conclusion of this paper is arbitrary.
AUTHORS: We did not claim in the text for any causal relationship between olfactory bulb changes in aging and behavior; however, we highlighted previously unrecognized correlations between in vivo detectable structural changes of the olfactory bulb and behavioral traits, which may ground future mechanistic studies.
There are still some minor concerns that should be addressed by the author.
Line 298: I recommend the author add a brief conclusion after the MRI result.
AU Done
Line 307: The author said the P value of the cord test is P<0.05, I recommend the author write the actual P value, P=0.038, which is the same as the number in Table 2. Replace all the P values with the actual P value that showed in Table 2 in this paper.
AU Done
Line 334: Please replot figure 2 and add a P value and * in each panel.
AU Asterisks (*) have been added as necessary, however writing the actual p values in the graphs with readable letters would complicate the figure too much. Actual values are reported in the text and Table 2, and they are duly referenced in the figure legend.
Line 334: The variation of the YOUNG group in Figure 2A looks too big. please check the P value again if you ignore the biggest value.
AU Carrying out the test without the bigger value yielded a P value = 0.05. We underscore that outliers may happen in every behavioral test because are part of the normal variability within a population at every age.
Line 387: Please replace “DL, light-to-dark” with “DL, dark-to-light”.
AU Done
Line 397: Please add a brief conclusion after this part.
AU Done
Reviewer 2 Report
This study demonstrated the behavioral changes in the aging mouse as well as the morphometric changes of the aging olfactory bulb. The authors showed adequate interesting data in this study, and this study appears to contribute to the related researchers.
I have one concern for this study. The authors found that FA in aging olfactory bulb (OB) was higher than that in young OB. The authors described the possible reason for this higher rate as “Coherently, an increased amount of myelin would justify for the measured increase in FA of the OB in elderly mice [50]” in lines 410~411. However, in [50], the corpus callosum, not OB, was examined. the authors should show the increase of myelin in the aging OB in this study.
Author Response
This study demonstrated the behavioral changes in the aging mouse as well as the morphometric changes of the aging olfactory bulb. The authors showed adequate interesting data in this study, and this study appears to contribute to the related researchers.
I have one concern for this study. The authors found that FA in aging olfactory bulb (OB) was higher than that in young OB. The authors described the possible reason for this higher rate as “Coherently, an increased amount of myelin would justify for the measured increase in FA of the OB in elderly mice [50]” in lines 410~411. However, in [50], the corpus callosum, not OB, was examined. the authors should show the increase of myelin in the aging OB in this study.
AUTHORS: We thank the Reviewer for this comment, which prompted for a more comprehensive discussion of the implications of MRI findings (lines 416-448 of the revised paper). We would like to underline that this work was not aimed at investigating the histological or biochemical counterparts of MRI findings; instead, we were interested in detecting possible relationships between MRI and behavior in living animals, to prospectively allow multiple testing in the same living animal.
Reviewer 3 Report
Since olfactory dysfunction seems to be involved in age-related dementia, such as Alzheimer’s disease, it is essential to understand how olfactory structure/function changes in normal aging and how they affect olfactory behaviors. In this study, authors examined differences between 6- and 16-month-old mice in the volume and structural characteristics of the olfactory bulb using MRI along with their behavioral patterns using many different behavioral tests that can assess motor, olfactory, cognitive performances, and emotional aspects. They then evaluated MRI variables with behavioral outputs using a multivariate analysis. I have two specific comments below.
Despite no difference in the size and total blood volume of the bulb between the young and old mice, the authors found the old mice had shorter T2 and greater FA compared to the young ones, suggesting an increase in myelination or a decrease in gray/white matter volume ratio in the old mice. I wonder if the authors determined T1 and saw a consistent change if the shortening of T2 was caused by myelination. Also, it would be nice to discuss where the change in myelination or gray/white matter ratio may occur. Could the change occur on the lateral olfactory tract and/or feedback projections from the anterior commissure?
Differences in outcomes found in the behavioral tests seem to be attributed to factors other than mere motor/olfactory performance differences. Indeed, no motor and olfactory deficits were detected in the old mice (e.g., the old mice swam longer distances!). The most intriguing finding was the differences in the light avoidance test that might tell defense reaction strategies. I wonder if the sleep/wake states of the mice affect the outcomes of this behavioral test. Mice are active in the night and inactive or sleep in daylight. Was the light avoidance test performed in daylight or at night? Could older mice be more inactive in daylight and less active at night than young mice? Also, although I agree stress/emotion affects olfactory behavior, I had difficulty linking the outcomes of the light avoidance task and olfactory structural changes. There seems no olfactory cue in the light avoidance task.
Author Response
REV: Despite no difference in the size and total blood volume of the bulb between the young and old mice, the authors found the old mice had shorter T2 and greater FA compared to the young ones, suggesting an increase in myelination or a decrease in gray/white matter volume ratio in the old mice. I wonder if the authors determined T1 and saw a consistent change if the shortening of T2 was caused by myelination.
AUTHORS: Unfortunately, we did not measure T1 due to the need for limiting total acquisition time in fragile elderly mice.
Also, it would be nice to discuss where the change in myelination or gray/white matter ratio may occur. Could the change occur on the lateral olfactory tract and/or feedback projections from the anterior commissure?
AU The limited (see previous comment) in vivo acquisition time and the available resolution at 4.7T did not allow for such a subtle investigation.
R:Differences in outcomes found in the behavioral tests seem to be attributed to factors other than mere motor/olfactory performance differences. Indeed, no motor and olfactory deficits were detected in the old mice (e.g., the old mice swam longer distances!). The most intriguing finding was the differences in the light avoidance test that might tell defense reaction strategies. I wonder if the sleep/wake states of the mice affect the outcomes of this behavioral test. Mice are active in the night and inactive or sleep in daylight. Was the light avoidance test performed in daylight or at night? Could older mice be more inactive in daylight and less active at night than young mice? Also, although I agree stress/emotion affects olfactory behavior, I had difficulty linking the outcomes of the light avoidance task and olfactory structural changes. There seems no olfactory cue in the light avoidance task.
R: “Indeed, no motor and olfactory deficits were detected in the old mice (e.g., the old mice swam longer distances!).”
AU: We underlined this counterintuitive result to rule out a trivial explanation of the elderly behavior e.g., ‘old mice move less’: this is not true, the elderly mice here move around the pool much more than younger mice.
R: “The most intriguing finding was the differences in the light avoidance test that might tell defense reaction strategies. I wonder if the sleep/wake states of the mice affect the outcomes of this behavioral test.”
AU: We thank the Reviewer for this insightful suggestion. Sleep-wake cycle in mice is very different from humans and rats, since it is more fragmented and consists of brief bouts of activity scattered throughout the 24 hours, more frequent during the night (Weber and Dan, 2016 doi:10.1038/nature19773). As shown in Fig. 1D (Weber and Dan 2016), mice are not inactive at daytime, as it happens for rats. All the behavioral tests were done in the same time window (in our morning, between 9.30 and noon): the results of the swim test (older mice are more active) rule out that the behavioral results are simply due to a drowsiness period. Nevertheless, in our experiments we did not record basal motor activity or EEG patterns, so in the future it would be very interesting to add also these parameters to highlight a possible influence of sleep/wake cycles on the behavioral results.
R: “I had difficulty linking the outcomes of the light avoidance task and olfactory structural changes. There seems no olfactory cue in the light avoidance task.”
AU: The Reviewer is right. We performed this test since it is used to explore emotional reactivity in mice, which is managed by olfactory brain areas, and could add a piece of information in addition to the other behavioral tests. It should be seen in the context of the complex dataset emerging from all the behavioral data: an “altered” olfactory input (due to olfactory bulb changes in the elderly) to olfactory areas managing emotional reactivity (e.g., the amygdala), could result in different behavioral patterns. So, we were not exploring the effect of the presence of an actual olfactory input, but the possible disruption (or modification) in emotional behavior, stemming from OB modified olfactory inputs to downstream behavior-controlling areas (e.g., amygdala).
Reviewer 4 Report
This paper describes correlations between MRI-based parameters of the olfactory bulb (OB) of mice with age and behavior. The behavioral tests described (e.g., olfaction, motor function, cognition) and structural differences of the OB with age are already known. Thus, this work demonstrates that the reported changes in OB can be directly detected and related to age.
The strength of the ms is the detection of age- and behavior-related structural differences of the OB using noninvasive measurements and its correlation with age and behavior.
However, the correlations described here are subtle, and the reasons for altered MRI parameters, e.g., lower T2 in old mice, need invasive (histological) explanation anyway. The term "structural" seems somewhat stretched, since it refers only to volume and leaves much room for interpretation that cannot be afforded here.
After all, the paper is an elegant approach to non-invasive, but elaborate demonstration of the link between chemosensory “hardware” and behavioral aspects.
There are only some minor points:
Introduction: Pivotal paper of Doty RL et al. (1984) should be cited here. In contrast to other senses the olfactory decline starts late (around 60y).
Methods: The buried pellet test was used to asses food finding (2.2.5). To exclude possible biases related to motor deterioration with age, other authors recommend a “surface pellet test” where the pellet remains visible (unburied) (Lehmkuhl et al. 2014). Please add why you did not use this test.
To understand the morphologic (MRI) part of the study, it would be helpful to include images of the OB showing the structural basis of the measurements.
Although mentioned in the discussion, the caveats of this study need to be pointed out more clearly: Reasons for lower T2 are not clear (increased myelination or increasing loss of neurons?)
Author Response
REVIEWER: “This paper describes correlations between MRI-based parameters of the olfactory bulb (OB) of mice with age and behavior. The behavioral tests described (e.g., olfaction, motor function, cognition) and structural differences of the OB with age are already known. Thus, this work demonstrates that the reported changes in OB can be directly detected and related to age.
The strength of the ms is the detection of age- and behavior-related structural differences of the OB using noninvasive measurements and its correlation with age and behavior.
However, the correlations described here are subtle, and the reasons for altered MRI parameters, e.g., lower T2 in old mice, need invasive (histological) explanation anyway. The term "structural" seems somewhat stretched, since it refers only to volume and leaves much room for interpretation that cannot be afforded here.
After all, the paper is an elegant approach to non-invasive, but elaborate demonstration of the link between chemosensory “hardware” and behavioral aspects.
AUTHORS: We thank the reviewer for appreciating our work. Once more, we underline that the main aim of the work was to explore possible relationships between in vivo measurable MRI and behavioral variables.
There are only some minor points:
Introduction: Pivotal paper of Doty RL et al. (1984) should be cited here. In contrast to other senses the olfactory decline starts late (around 60y).
AU Done
Methods: The buried pellet test was used to asses food finding (2.2.5). To exclude possible biases related to motor deterioration with age, other authors recommend a “surface pellet test” where the pellet remains visible (unburied) (Lehmkuhl et al. 2014). Please add why you did not use this test.”
AU: At line 150 of the original manuscript, we wrote: “Then the test was repeated placing the food pellet in visible position over the sawdust, to control for motivation to eat.” Actually, in our laboratory we use such protocol (1. food pellet finding in invisible position, 2. food pellet in visible position) as a control condition since we started to use the Food-finding test, see for example: Massimino et al Physiology & Behavior 119 (2013) 86–91, for mice models of neurological diseases (at any age). This reference to our previous work has been included.
R: ”To understand the morphologic (MRI) part of the study, it would be helpful to include images of the OB showing the structural basis of the measurements.
Although mentioned in the discussion, the caveats of this study need to be pointed out more clearly: Reasons for lower T2 are not clear (increased myelination or increasing loss of neurons?)”
AU: A figure showing the MRI of representative OB in an adult mouse was added (New Fig. 2). The discussion on the counterparts of MRI changes was expanded (see answer to Reviewer 2).
Reviewer 5 Report
The authors P. Bontempi, MJ Ricatti, M Sandri, E Nicolato, C Mucignat-Caretta and C Zancanaro in their study “Age-related in-vivo structural changes in the male mouse olfactory bulb and olfactory-driven behavior” describe the differences in motor, olfactory, cognitive and emotional performance between young and elderly mice.
This is a group that has substantial experience in the investigation of the olfactory system, particularly with regard to its morphological and functional aspects. This is shown in the writing, which is agile, precise, and clear. It is also evident in the thoroughness with which the methodology of the statistical analysis was executed and the results were interpreted.
Unfortunately, the results are not particularly revealing or significant in the sense that the significant differences in motor behavior and olfactory sensitivity are reduced to a very small number of tests, which sometimes offer counter-intuitive or contradictory results. Similarly, the findings of the MRI do not provide a lot of new information. Because of this, it is prudent to proceed with extreme caution when attempting to establish a connection between the concurrent occurrence of age-related behavioral changes and anatomical changes in the olfactory bulb.
On the other hand, it would have been worthwhile to conduct postmortem histological research on some of the specimens with the most extreme MRI findings in order to confirm if the postulated cause of the differences in T2 and FA was a rise in the white matter or a decrease in the gray matter.
Minor issues:
1) From reading the title it is not clear whether the authors are looking at the effects of aging on both bulb structure and animal behavior or whether they are looking at how changes in bulb structure affect olfactory-mediated behavior. Reading the paper clarifies this aspect, but perhaps it would be helpful to find an alternative and less confusing rewording of the title.
2) Line 107-108: MRI has been used to investigate the structure and function of the olfactory system [31-39]. MRI has been shown amenable to studying the mouse olfactory bulb in detail [39].
Both sentences should be joined.
3) This references of olfactory bulb and aging should be relevant for the study:
Mirich JM, Williams NC, Berlau DJ, Brunjes PC. Comparative study of aging in the mouse olfactory bulb. J Comp Neurol. 2002 Dec 23;454(4):361-72. doi: 10.1002/cne.10426. PMID: 12455003.
Forbes WB. Aging-related morphological changes in the main olfactory bulb of the Fischer 344 rat. Neurobiol Aging. 1984 Summer;5(2):93-9. doi: 10.1016/0197-4580(84)90037-x. PMID: 6493438; PMCID: PMC7134915.
Author Response
REVIEWER: “The authors P. Bontempi, MJ Ricatti, M Sandri, E Nicolato, C Mucignat-Caretta and C Zancanaro in their study “Age-related in-vivo structural changes in the male mouse olfactory bulb and olfactory-driven behavior” describe the differences in motor, olfactory, cognitive and emotional performance between young and elderly mice. This is a group that has substantial experience in the investigation of the olfactory system, particularly with regard to its morphological and functional aspects. This is shown in the writing, which is agile, precise, and clear. It is also evident in the thoroughness with which the methodology of the statistical analysis was executed and the results were interpreted. Unfortunately, the results are not particularly revealing or significant in the sense that the significant differences in motor behavior and olfactory sensitivity are reduced to a very small number of tests, which sometimes offer counter-intuitive or contradictory results. Similarly, the findings of the MRI do not provide a lot of new information. Because of this, it is prudent to proceed with extreme caution when attempting to establish a connection between the concurrent occurrence of age-related behavioral changes and anatomical changes in the olfactory bulb.
AUTHORS: The reviewer is right. Accordingly, we avoided suggesting any causative role and only claimed for “correlation” between MRI variables and behavior, and “suggested” possible explanations for age-related MRI changes.
R On the other hand, it would have been worthwhile to conduct postmortem histological research on some of the specimens with the most extreme MRI findings in order to confirm if the postulated cause of the differences in T2 and FA was a rise in the white matter or a decrease in the gray matter.
AU The goal of the present work was to determine a protocol for (possibly repeated) testing with MRI and behavioral tests during aging. Please see answers to Reviewer 2 and 4. A detailed histological examination will be carried out in the future, with a more targeted experiment (selected behavioral experiments and more focused MRI).
R Minor issues:
- From reading the title it is not clear whether the authors are looking at the effects of aging on both bulb structure and animal behavior or whether they are looking at how changes in bulb structure affect olfactory-mediated behavior. Reading the paper clarifies this aspect, but perhaps it would be helpful to find an alternative and less confusing rewording of the title.
AU Done
R 2) Line 107-108: MRI has been used to investigate the structure and function of the olfactory system [31-39]. MRI has been shown amenable to studying the mouse olfactory bulb in detail [39].
Both sentences should be joined.
AU Done
R 3) This references of olfactory bulb and aging should be relevant for the study:
Mirich JM, Williams NC, Berlau DJ, Brunjes PC. Comparative study of aging in the mouse olfactory bulb. J Comp Neurol. 2002 Dec 23;454(4):361-72. doi: 10.1002/cne.10426. PMID: 12455003.
Forbes WB. Aging-related morphological changes in the main olfactory bulb of the Fischer 344 rat. Neurobiol Aging. 1984 Summer;5(2):93-9. doi: 10.1016/0197-4580(84)90037-x. PMID: 6493438; PMCID: PMC7134915.”
AU: We thank the Reviewer for the suggestion, both references are relevant and have been added to the refence list.
Round 2
Reviewer 3 Report
Thank you for addressing my concerns. I do not have further questions.